# Overexpression of microRNA-21-5p and microRNA-221-5p in Monocytes Increases the Risk of Developing Coronary Artery Disease

**DOI:** 10.3390/ijms24108641

**Published:** 2023-05-12

**Authors:** Yazmín Estela Torres-Paz, Ricardo Gamboa, Giovanny Fuentevilla-Álvarez, María Elena Soto, Nadia González-Moyotl, Rocío Martínez-Alvarado, Margarita Torres-Tamayo, Edgar Samuel Ramírez-Marroquín, Xicoténcatl Vásquez-Jiménez, Víctor Sainz-Escarrega, Claudia Huesca-Gómez

**Affiliations:** 1Physiology Department, Instituto Nacional de Cardiología “Ignacio Chávez”, México City 14080, Mexicorgamboaa_2000@yahoo.com (R.G.); fuentevilla_alvarez@hotmail.com (G.F.-Á.); nadigm93@gmail.com (N.G.-M.); 2Postgraduate Program in Medical, Dental and Health Sciences, Universidad Nacional Autónoma de México (UNAM), México City 04510, Mexico; 3Biochemistry Department, Escuela Nacional de Ciencias Biológicas, Instituto Politécnico Nacional (IPN), México City 11350, Mexico; 4Immunology Department, Instituto Nacional de Cardiología “Ignacio Chávez”, México City 14080, Mexico; mesoto50@hotmail.com; 5Master’s Program in Health Science, Escuela Superior de Medicina, Instituto Politécnico Nacional (IPN), México City 11350, Mexico; 6Endocrinology Department, Instituto Nacional de Cardiología “Ignacio Chávez”, México City 14080, Mexico; orssino@yahoo.com (R.M.-A.); mtt1199@yahoo.com (M.T.-T.); 7Cardiothoracic Surgery Department, Instituto Nacional de Cardiología “Ignacio Chávez”, México City 14080, Mexico; samuelramirez50@gmail.com (E.S.R.-M.); dr.xicovasquez@hotmail.com (X.V.-J.); victorsainze@gmail.com (V.S.-E.)

**Keywords:** microRNA, monocytes, coronary artery disease

## Abstract

MicroRNAs (miRs) regulate gene expression at the post-transcriptional level and are found to be present in monocytes. This study aimed to investigate miR-221-5p, miR-21-5p, and miR-155-5p, their expression in monocytes, and their role in coronary arterial disease (CAD). The study population comprised 110 subjects, and RT-qPCR was used to examine the miR-221-5p, miR-21-5p, and miR-155-5p expressions in monocytes. Results: the miR-21-5p (*p* = 0.001) and miR-221-5p (*p* < 0.001) expression levels were significantly higher in the CAD group, and the miR-155-5p (*p* = 0.021) expression levels were significantly lower in the CAD group; only miR-21-5p and miR-221-5p upregulation was found to be associated with an increased CAD risk. The results show significant increases in miR-21-5p in the unmedicated CAD group with the metformin patients vs. the healthy control group (*p* = 0.001) and vs. the medicated CAD group with metformin (*p* = 0.022). The same was true for miR-221-5p in the CAD patients unmedicated with metformin vs. the healthy control group (*p* < 0.001). Our results from Mexican CAD patients show that the overexpression in monocytes of miR-21-5p and miR-221-5p increases the risk of the development of CAD. In addition, in the CAD group, the metformin downregulated the expression of miR-21-5p and miR-221-5p. Also, the expression of endothelial nitric oxide synthase (NOS3) decreased significantly in our patients with CAD, regardless of whether they were medicated. Therefore, our findings allow for the proposal of new therapeutic strategies for the diagnosis and prognosis of CAD and the evaluation of treatment efficacy.

## 1. Introduction

Coronary arterial disease (CAD) is the pathologic process of atherosclerosis affecting the coronary arteries and is considered the most common type of heart disease [1]. The main cardiovascular risk factor is the formation of atherosclerotic plaque [2,3]. The process begins with the reactive oxygen species’ (ROS) overproduction in the intima layer of the coronary artery, an increase in vasoconstrictors, and a decrease in vasodilation agents, which is known as endothelial dysfunction [4]. This damage causes greater endothelial permeability, allowing the passage of low-density lipoproteins (LDL) to the intima, where they are oxidized by ROS [5]. Nitric oxide (NO) is the main vasodilator affected during the atherosclerotic process. Endogenous production is dependent on the activity of the enzyme endothelial NO synthase (eNOS) [6]. The bioavailability of NO is affected due to an increase in ROS, which is a hallmark feature of endothelial dysfunction in vascular disease states [7]. According to the literature, the expression of eNOS (NOS3) and the NO release can be regulated post-transcriptionally by small non-coding RNAs (miRs) [8].

The recruitment and internalization of circulating monocytes into the arterial intima constitutes one of the first steps in the initiation and progression of atherosclerosis [6]. Monocytes have been identified as expressing microRNAs (miRs). MiRNAs are small non-coding RNA sequences that are present in the genomes of a great variety of organisms and that bind to the 3′ untranslated region (UTR) of specific mRNAs according to the complementarity of their sequences. As a result of this bind, miRNAs can inhibit translation or promote mRNA degradation [9,10], and are consequently involved in the regulation of gene expression at the post-transcriptional level. Moreover, they are considered molecular markers for the detection of various diseases because of their presence in various body fluids, such as serum and plasma. Some of the miRs have been found to be involved in inflammation and oxidative stress, and their dysregulation has been implicated in various diseases, including cardiovascular diseases and atherosclerotic plaque formation [11].

Some of the miRs found to be involved in inflammation and oxidative stress—processes occurring during the pathogenesis of atherosclerosis—include miR-21-5p, miR-155-5p, and miR-221-5p. MiR-21-5p has been described as being highly expressed in different cell types of the cardiovascular system; however, under cardiovascular disease conditions, its expression is deregulated in the heart and vasculature [12]. MiR-21 is abundant in the vessel wall and responds with differential expression upon shear or mechanical stress to the vessel. In humans, miR-21 is expressed in most cell types but is highly expressed in podocytes, dendritic cells, and CD14+ monocytes [13]. MiRNA-21 expression on endothelial cells has been found to be significantly upregulated by shear stress treatment and to cause an anti-apoptotic effect by directly targeting the PTEN tumor suppressor gene [14]; it’s overexpression inhibits the expression of the peroxisome proliferator-activated receptor-alpha (PPARα), resulting in the upregulation of the expressions of VCAM-1 and MCP-1 [15].

On the other hand, it is known that miR-221-3p regulates different vascular physiological processes and is involved in various vascular-related pathological mechanisms [16]. Nevertheless, in recent years, the 5p chain of this miR has been associated with left ventricular reverse remodeling, heart failure [17], and arteriosclerosis obliterans [18]. Another miR related to the inflammation process is miR-155-5p; its expression is affected by various inflammatory signals, including lipopolysaccharide, interferon-β, tumor necrosis factor, and the atheromatous process [19,20]. Hematopoietic miR-155-5p deficiency in hyperlipidemic mice increases the development of atherosclerotic plaques, thus decreasing their stability [21]. In previous studies, the analysis of monocyte miRNA expression profiles has been based on identifying differences in expression between different subsets of monocytes [22,23]. However, little is known about the role of miRs on monocytes in atherosclerotic plaque formation. Thus, in the present study, we aimed to characterize the expression of miR-21-5p, miR-155-5p, and miR-221-5p in the monocytes of CAD patients and elucidate the details of their role in this disease.

## 2. Results

### 2.1. Characteristics of the Study Population

The study population consisted of 110 Mexicans recruited at the National Cardiology Institute Ignacio Chávez: 50 patients were in the healthy control group (CG), and 60 patients were in the CAD group. Table 1 shows the main biochemical and anthropometric parameters of these groups. The CAD group shows significantly higher values of age (*p* < 0.001), glucose (*p* < 0.001), systolic blood pressure (SBP) (*p* < 0.001), diastolic blood pressure (DBP) (*p* < 0.001), and heart rate (*p* < 0.001), and significantly lower values of CT (*p* < 0.001) and HDL-C (*p* < 0.001). A 43% prevalence of diabetes was reported among the CAD patients, and 55% of them had arterial hypertension (HT).

### 2.2. miR Expression in Monocytes

MiR expression in monocytes was significantly higher in the CAD group compared with the control group (CG) in the case of miR-21-5p (median: 15.13 (min. 1.45–max. 130.43) vs. median: 7.54 (min. 0.69–max. 61.42); *p* = 0.001, respectively) (Figure 1A), as well as in the case of miR-221-5p (median: 21.12 (min. 1.67–max. 116.41) vs. median: 7.47 (min. 0.12–max. 74.64); *p* < 0.001, respectively) (Figure 1B). On the contrary, miR-155-5p expression was significantly lower in the CAD group compared with the CG (median: 7.69 (min. 0.91–max. 33.15) vs. median: 14.04 (min. 0.74–max. 78.72); *p* = 0.021, respectively) (Figure 1C).

### 2.3. NOS3 Expression in Monocytes

NOS3 is responsible for the synthesis of nitric oxide in blood vessels, and its mRNA is targeted by miR-21-5p, miR-155-5p, and miR-221-5p [24,25]. The binding of multiple miRNAs to the same mRNA through different target sites can have a considerable effect on gene function [25]. Therefore, we also determined the expression of NOS3 mRNA in the monocytes. Its expression was significantly lower in the CAD group compared to the CG (median: 27.01 (min. 1.00–max. 129.55) vs. median: 108.44 (min. 5.70–max. 1125.22); *p* < 0.001, respectively) (Figure 1D).

### 2.4. Association of Clinical Parameters and miR/NOS3 Expression with the Risk of Developing CAD

Binary logistic regression analysis was used to identify whether the deregulation of miR expression influences the development of CAD using the clinical parameters and miRs that were shown to be statistically significantly different between the control group and the CAD group (sex, age, total cholesterol, HDL-C, glucose, SBP, DBP, miR-21-5p, miR-155-5p, miR-221-5p, and NOS3). Table 2 shows that increased miR-21-5p and miR-221-5p expression and decreased NOS3 expression are associated with a higher risk of CAD; conversely, HDL-C is a protective factor against CAD. We also found that increasing age and SBP are weakly associated with CAD.

### 2.5. miR/NOS3 Expression with the Use of Drugs

Because most of the CAD patients were medicated with different drugs, a Kruskal–Wallis test was performed between the CG and medicated and unmedicated CAD patients to determine whether the expression of the studied miRs/NOS3 varies according to the use of metformin, statins, or antihypertensive drugs. NOS3 mRNA expression showed an increase in the control group versus the medicated and unmedicated CAD patients with metformin. MiR-21-5p expression levels were higher in the unmedicated CAD patients with metformin compared with the healthy control group (*p* = 0.001) and in the CAD medicated group with metformin (*p* = 0.022) (Table 3). Moreover, miR-221-5p expression in the unmedicated CAD patients with metformin was significantly higher than that in the healthy control group (*p* < 0.001), and it showed a tendency to be higher than that in the hypoglycemic-medicated patients (*p* = 0.086). We found no differences in the expression of miRs or NOS3 between the medicated and unmedicated groups due to the consumption of statins and antihypertensive drugs.

## 3. Discussion

Several studies have indicated the potential role of circulating miRNA levels as valuable biomarkers for different disease processes. However, the information is limited regarding their role in monocytes, cells that are critical in the initial stage of atherosclerosis.

In our study, we demonstrated that monocytes from CAD patients increased the expression of miR-21-5p and miR-221-5p and decreased the expression of miR-155-5p and NOS3. However, only the overexpression of miR-21-5p and miR-221-5p was found to be associated with an increased CAD risk. Previous reports have shown that the expression of miR-21-5p is elevated in the mononuclear cells and plasma of CAD patients [24,25]. This miR could potentially be involved by regulating inflammatory cytokines through the JNK signaling pathway [26] or the phosphatidylinositol 3-hydroxy kinase [PI3K]/protein kinase B [AKT] pathway [27], which would cause vascular smooth muscle cell proliferation and migration, resulting in more severe vascular stenosis and worsening CAD [28]. According to the literature, miRNA-21 overexpression inhibits the expression of the peroxisome proliferator-activated receptor-alpha (PPARα), resulting in the upregulation of the expressions of VCAM-1 and MCP-1 [15]. Furthermore, hydrogen peroxide and lipopolysaccharide (LPS) have been found to differentially affect the expression of miR-21 in endothelial cells before and after co-culture with monocytes [29]. These studies demonstrate the role of miR-21 in the development of the atherosclerosis process; however, the effect of increased miR-21 expression in endothelial cells in the context of atherosclerosis requires further examination.

Some researchers have reported that miR-221-5p expression is upregulated in plasma from patients with heart failure and atherosclerosis obliterans [17,30]. In addition, studies in endothelial cells have shown that the expression of the 3p chain of this miR is elevated in patients with CAD [15,31]. However, there are no previous reports of the 5p chain in this disease. Therefore, the present study is the first to demonstrate that an increase in miR-221-5p in monocytes could facilitate the formation of atheromatous plaque. We used the network-based bioinformatic prediction TargetScan Human 7.2 program and identified the suppressor of the cytokine signaling 1 (SOCS1) gene as a direct anti-inflammatory target of the miR-221-5p. SOCS1 is a negative feedback inhibitor of signaling induced by the cytokines that act via the JAK/STAT pathway, and it controls the essential inflammatory processes in vascular cells [32]. Hence, the overexpression of miR-221-5p could promote the production of inflammatory cytokines and the progression of atherosclerosis through the inhibition of SOCS1.

CAD management is affected by associated pathologies such as diabetes mellitus (DM), which results in a two to four times higher CVD risk. It should be noted that because a high percentage of our patients with CAD had type 2 diabetes mellitus and arterial hypertension, most received medication for these. Therefore, to verify that the expression of these miRs is directly associated with CAD, a binary logistic regression analysis was performed; it was confirmed that their deregulation, either up or down, is a risk factor for developing the disease (as shown in the Table 2). If the deregulation in the expression of these was due to other comorbidities—in this case type 2 diabetes mellitus or hypertension—the expression of the miRNAs and NOS3 would not have given us a significant result in the regression, and this would have indicated that the deregulation observed in its expression was not directly associated with CAD but rather with one of these comorbidities or other confounding variables. 

However, it is difficult to separate the effects of diabetes, arterial hypertension, and medication from the effects of CAD on miRNA expression and NOS3 levels. It is necessary to consider the difficulty in finding patients with CAD who are not treated, and thus the need to increase the sample size of this type of patient. A follow-up study could be done on our study subjects to (1) Determine whether healthy controls that presented a high expression of miRNA-21-5p and miRNA-221-5p develop CAD in the following years. Conversely, to determine whether those healthy subjects who presented a high expression of miR-NA-155-5p and NOS3 do not present with CAD (protective effect). (2) To determine whether patients with CAD who presented with a high expression of miRNA-21-5p and miRNA-221-5p have complications resulting from the disease or a poor prognosis.

According to the results of the treatment with the metformin, the patients with CAD who did not receive the medication have higher expression of miR-21-5p and miR-221-5p than both the control group and the group of CAD patients medicated with metformin. Previous reports have shown that treatment with metformin decreases the expression of both miR-21-5p in plasma and miR-221-5p in internal mammary arteries in diabetic patients [33,34]. This suggests that metformin also has an inhibitory effect on miR-21-5p and miR-221-5p expression in monocytes. Metformin acts as a cardioprotector by activating the AMPK (AMP-activated protein kinase) enzyme complex, which is involved in a wide range of downstream signaling pathways; participating in the apoptosis of endothelial cells and cardiomyocytes; inhibiting the proliferation, migration, and angiogenesis of vascular cells; and in mitigating oxidative stress and the inflammatory response [35,36]. Pulito et al. found that metformin inhibits miR-21-5p transcription through the occupancy of its promoter regions by E2F transcription factor 3 (E2F3) [37]. Nevertheless, the molecular mechanism of metformin in regulating the expression of the studied miRs is still not fully known.

No significant differences were found with respect to miR-221-5p, miR-21-5p, miR-155-5p, and NOS3 expressions grouped by CAD patients and their consumption of different antihypertensive and lipid-lowering drugs. This result is likely due to the wide variety of treatments involving antihypertensive drugs (Diuretics, Beta-blockers, ACE inhibitors, Angiotensin II receptor blockers, Calcium channel blockers, Alpha blockers, Alpha-2 Receptor Agonists, Vasodilators) and lipid-lowering drugs (statins, Fibrate, ezetimibe) in patients with CAD. 

In our study, the miR-155-5p was also shown to be downregulated in the CAD patients. It has been suggested in the literature that miR-155-5p expression may have different effects depending on the stage of atheromatous plaque development. Its downregulation could represent a feedback mechanism for controlling the overactivation of immune cells. According to a previous mouse model study with atherosclerosis, the expression of miRNA-155-5p decreases in these cells in response to inflammation, and thus it could be associated with the formation of atherosclerotic plaque [38]. 

Therefore, we recognize that miR-155-5p expression decreases in the monocytes of CAD patients in response to an increase in inflammatory cytokines caused by immune cell activation. McCoy et al. proposed a possible mechanism, showing that IL-10, a potent anti-inflammatory cytokine, inhibits the transcription of miR-155-5p of the BIC gene in a STAT3-dependent manner, which could result in the promoted expression of anti-inflammatory genes [39]. Zhu et al. demonstrated that miR-155-5p could have an anti-inflammatory effect by targeting mitogen-activated protein kinase 10 (MAP3K10), preventing the development and progression of atherosclerosis [19]. Studies on rodents demonstrate that the lack of miR-155-5p in bone marrow-derived cells increased the incidence of atherosclerosis in LDLR-KO mice [40]. Moreover, another study found that miR-155-5p partially mediated the effect of inflammatory stimuli (TNFα) induced endothelial nitric oxide synthase downregulation in HUVEC cells [41].

We observed that the decrease in miR-155-5p was even more significant among CAD patients receiving metformin treatment. The reduction of miR-155-5p in the B cells of subjects with DM who take metformin has been found to cause AMPK activation and, consequently, the inhibition of the pro-inflammatory cytokine TNFα [21]. Furthermore, it is known that miR-155-5p is induced by ox-LDL and proinflammatory stimuli in the monocyte/macrophages, which promotes a proinflammatory environment in which various cytokines are triggered [42]. However, most of our patients with CAD had low LDL levels due to atorvastatin and low susceptibility to oxidation. Studies show that simvastatin pretreatment ameliorates TNFα (20 ng/mL) induced miR-155-5p expression in HUVECs, further clarifying how statin modulation of the mevalonate-geranylgeranyl pyrophosphate-RhoA signaling pathway has anti-atherosclerotic effects [43]. Therefore, it is necessary, for example, to carry out complementary studies in vivo to ensure that its inhibition in mice with atherosclerosis produces a decrease in the size of the plaque. 

Furthermore, we observed a significant decrease in NOS3 expression in the monocytes of the CAD group. A previous study in mice demonstrated that NOS3 deficiency increased atherosclerotic plaque formation and induced coronary artery disease and several cardiovascular complications, including spontaneous aortic aneurysm and dissection [44]. In addition, a study of hypertensive patients indicated that miR-21-5p expression was positively correlated with cIMT and negatively correlated with serum nitric oxide concentrations and plasma activity NOS3 [24], as in our study. Previous studies are related the expression of miRNA-21-5p and miRNA-221-5p with NOS3 in both animal models and in humans [45,46]. Recent reports have also demonstrated that miR-21-5p is induced during hypoxia by HIF-1-activating Akt, which could indirectly affect NO levels [47]. Studies have also indicated that miR-221-3p inhibits adiponectin-stimulated NOS3 phosphorylation and NO production [48], which supports the inverse correlation we found between the levels of expression of miR221 and NOS3. Cerda et al. demonstrated that endothelial cells that were transfected showed a decrease in miR-221 after statin treatment that was correlated with the increase in NOS3 mRNA levels [49]. On the other hand, Suárez et al. observed that the transfection of miRNAs mimicked in endothelial cells decreased the protein levels of NOS3 [50]. In the case of miRNA-21-5p, a significant increase in the expression of miRNA-21-5p correlated positively with CIMT and negatively with the expression of NOS3 and nitric oxide was found in hypertensive patients with a high intima media thickness [24].

In summary, our results show that the overexpression of miR-21-5p and miR-221-5p in monocytes is a risk factor in the development of CAD in the Mexican population. Hence, miR-21-5p and miR-221-5p could both participate in the development of atherosclerotic plaques by inhibiting target genes that participate in this pathological process. Moreover, metformin could exert a protective effect by decreasing the expression of miR-21-5p and miR-221-5p. Further, the expression of NOS3 decreased significantly in our patients with CAD, regardless of whether they were medicated. Our work verifies the role of these miRs in the Mexican population, whose genetic load is different from the Caucasian population, on whom most such studies have been carried out. Therefore, our findings allow for the suggestion of new therapeutic strategies for the diagnosis and prognosis of CAD and the evaluation of treatment efficacy. 

One limitation of our study was cardiovascular treatment, as it varied among patients, and most patients were treated with two or more medications. However, the recruitment of patients without any type of treatment was very difficult. The effect of metformin in reducing miRNA expression should be investigated further in vivo or in vitro in other models. Therefore, more studies with larger sample sizes can help us elucidate the association of different miRs in CAD patients with different medical treatments. It would be interesting to study patients with subclinical atherosclerosis and CAD patients who have had the disease for different numbers of years, to determine whether the expression of miRs increases or decreases as the disease progresses.

## 4. Materials and Methods

### 4.1. Research Population

A total of 110 Mexican subjects (60 patients diagnosed with CAD and 50 control subjects) were recruited at the National Institute of Cardiology Ignacio Chávez. The inclusion criteria for both groups were Mexican by birth and having at least three previous generations of Mexican origins. CAD was defined as a disease that causes stress- or exercise-related symptoms of angina due to a narrowing of ≥50% in the left common trunk or ≥70% in one or more of the major arteries. The diagnosis as non-invasive was based on the presence of myocardial ischemia in patients with atheromatous plaques. The criterion for the control group was not presenting with any comorbidity; to ensure that the patients in the control group did not have atheroma or subclinical atherosclerosis, their carotid intima-media thickness (cIMT) was evaluated using ultrasonography. All the participants answered standardized and validated questionnaires to obtain information on their family and medical history, alcohol and tobacco consumption, and physical activity. Information was also obtained from the clinical record of each subject. The research protocol 18–1075 was approved by the Institute’s Research and Ethics Committees. Informed consent to participate was signed by all the patients, with full consideration for the ethical principles for medical research involving human beings, as stipulated in the Declaration of Helsinki and modified by the Tokyo Congress, Japan [51].

### 4.2. Blood and Plasma Samples

Blood samples were obtained by venipuncture after 12 h of fasting in tubes with ethylenediaminetetraacetic acid sodium (EDTA-Na) for plasma and monocyte isolation. Plasma was immediately subjected to separation through centrifugation for the determination of the lipid profile total cholesterol (TC), triglycerides, high-density lipoprotein cholesterol (HDL-C), low-density lipoprotein cholesterol (LDL-C), and glucose.

### 4.3. Laboratory Analysis

Glucose, TC, and triglycerides were analyzed using enzymatic colorimetric methods (Roche-Syntex/Boheringer Mannhein, Mannheim, Germany). HDL-C was measured after the precipitation of low-density and very low-density lipoproteins with phosphotungstate/Mg2+ (Roche-Syntex), and LDL-C was estimated by the equation of Friedewald [52], with the modifications detailed by De Long [53]. All the assays were performed according to an external quality control scheme (Lipid Standardization Program, Center for Disease Control in Atlanta, GA, USA). We followed the National Cholesterol Education Project (NCEP) Adult Treatment Panel (ATP III) guidelines and thus defined dyslipidemia according to the following levels: cholesterol ≥ 200 mg/dL; LDL-C ≥ 130 mg/dL; HDL-C < 40 mg/dL for men and <50 mg/dL for women; and triglyceride ≥ 150 mg/dL.

### 4.4. Monocyte Isolation

Whole blood collected in EDTA tubes was diluted 1:1 with 1X PBS–1% heparin and subsequently added to a Ficoll-Histopaque solution (Sigma, Sigma, St. Louis, MO, USA, Catalog Number 10771). The samples were centrifuged at 1300 rpm for 30 min, and the peripheral blood mononuclear cells (PBMCs) were generated after the centrifugation. Then, positive selection to obtain monocytes was carried out using CD14-mAb-coated microbeads (Miltenyi Biotec, Bergisch Gladbach, Germany; Catalog Number 130-050-201) following the manufacturer’s instructions (purity of 95 ± 98%). An aliquot of 1 mL of Tripure TM reagent (Roche Life Science, Penzberg, Germany) was added for collecting the monocytes. The cells were stored at −80 °C.

### 4.5. RNA Extraction 

The total RNA, including miRs, was extracted from the monocyte samples using Tripure™ isolation reagent (Roche Diagnostics, Indianapolis, IN, USA, Catalog Number 11667165001). The total RNA was stored at −80 °C.

### 4.6. miR Quantitative Real-Time

Reverse transcription was performed using the samples of the total RNA from the monocytes to obtain the cDNAs of the study miRs using specific primers for the mature forms of each one and the TaqMan miRNA RT kit (TaqMan^®^ Advanced miRNA cDNA Synthesis Kit, Applied Biosystem, Foster City, CA, USA, Catalog Number A28007). The quantification of the miR-21-5p (hsa-miR-21-5p), miR-155-5p (hsa-miR-155-5p), and miR-221-5p (hsa-miR-221-5p) via a real-time quantitative reverse transcription polymerase chain reaction (qRT-PCR) was performed using a commercial kit (TaqMan Gene Expression Assay, Applied Biosystem, Foster City, CA, USA) employing the CFX96TM Touch Real-Time PCR Detection System (Bio-Rad, Hercules, CA, USA). The cycling conditions were 2 min at 50 °C and then 10 min at 95 °C, followed by 40 cycles of 15 s at 95 °C and 1 min at 60 °C. The expression levels were measured in duplicate and normalized to the reference gene RNU6B (NR_002752). The relative expression was calculated with the comparative threshold cycle (CT) method, and the data were analyzed using the 2^−ΔΔCt^ method [54].

### 4.7. mRNA Quantitative Real-Time PCR

The TR-qPCR was performed using 1 µg of the total RNA for cDNA synthesis according to the High-Capacity cDNA Reverse Transcription kit (Applied Biosystem, Foster City, CA, USA). The cDNA was stored at −20 °C. NOS3 (Hs01574665_m1) and was measured using a commercially available kit (TaqMan Gene Expression Assay, Applied Biosystem, Foster City, CA, USA) employing the CFX96TM Touch Real-Time PCR Detection System (Bio-Rad, Hercules, CA, USA). The cycling conditions were 2 min at 50 °C and 10 min at 95 °C, followed by 40 cycles of 15 s at 95 °C and 1 min at 60 °C. The gene expression levels were determined in duplicate and normalized with the reference gene HPRT (Hs99999909_m1).

### 4.8. Statistical Analysis

The data were analyzed using SPSS version 22 (SPSS Inc., Chicago, IL, USA). A descriptive analysis of all the variables was performed, and the results were expressed as the mean ± standard deviation (SD). The comparison between the groups was rendered using Student’s t-test for continuous variables. The Mann–Whitney U test was performed for the variables that did not present a normal distribution. The Kruskal–Wallis test was performed using a comparison of the medians to evaluate the outcomes of the control group, the unmedicated CAD patients, and the CAD patients medicated with statins or antihypertensive and hypoglycemic agents. Subsequently, a binary logistic regression analysis was conducted to explore the association between the study miRs and the CAD. Finally, a correlation analysis was conducted using the Pearson test. The *p* values of <0.05 were considered as indicating significant differences.

### 4.9. Sample Size

The sample size was calculated using the data published in recent years by various researchers. The means and standard deviations in the data were considered. The calculated size of the present study was 28 subjects [19]; however, a sample size of 50 subjects per group was established.
$$n=\frac{{\left(Z_{1-\alpha /2}+Z_{1-\beta }\right)}^{2}\left(S_{1}^{2}+S_{2}^{2}\right)}{{\left(X_{1}-X_{2}\right)}^{2}}$$

(Z_1−𝛼/2_) = 1.96 y 2.576 for a confidence level of 95% and 99%, respectively

(Z_1−𝛽_) = 0.84 y 2.326 for a statistical power of 80% and 99%, respectively

*S*_1_ = Standard deviation of the control group

*S*_2_ = Standard deviation of the group problem

*X*_1_ = Control of population means of the group

*X*_2_ = Population mean of the problem group.

## Figures and Tables

**Figure 1 ijms-24-08641-f001:**
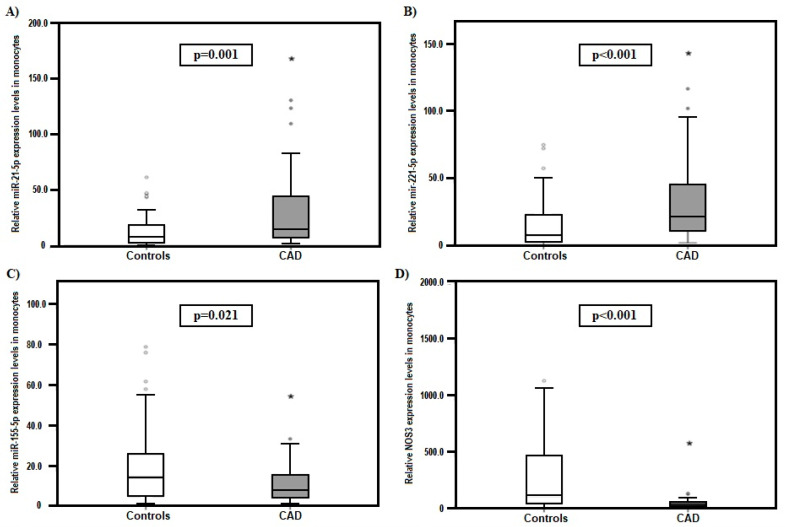
Comparison between CG and CAD patients’ expression levels in monocytes of (**A**) miR-21-5p, (**B**) miR-221-5p, and (**C**) miR-155-5p, (**D**) *NOS3* expression. The data were normalized to RNU6B. The data are expressed as medians (min.–max.) (Mann–Whitney U test). * *p* > 0.05.

**Table 1 ijms-24-08641-t001:** Biochemical and anthropometric parameters of the study population.

Variable	Controls (n = 50)	CAD (n = 60)	*p*
Age (years)	48.67 ± 6.47	62.10 ± 11.75	<0.001
Gender % M/W	60.7%/39.3%	85.3%/14.7%	**0.001**
BMI (kg/m^2^)	27.75 ± 3.78	26.42 ± 3.90	0.849
Total cholesterol (mg/dL)	171.06 ± 24.47	138.26 ± 49.98	**<0.001**
HDL-C (mg/dL)	45.80 ± 14.75	30.98 ± 7.90	**<0.001**
LDL-C (mg/dL)	102.86 ± 23.17	90.61 ± 49.70	0.066
Triglycerides (mg/dL)	131.48 ± 49.86	141.94 ± 62.30	0.302
Statins %	0%	72.0%	**<0.001**
Glucose (mg/dL)	93.76 ± 7.82	120.34 ± 48.78	**<0.001**
Diabetes %	0%	43.0%	**<0.001**
Hypoglycemic agents %	0%	20.0%	**<0.001**
SBP (mmHg)	112.43 ± 9.12	127.33 ± 18.81	**<0.001**
DBP (mmHg)	70.43 ± 6.01	79.60 ± 12.03	**<0.001**
Hypertension %	0%	55.0%	**<0.001**
CF (pulsations/min)	65.44 ± 9.81	79.44 ± 13.10	**<0.001**
Antihypertensive %	0%	62.0%	**<0.001**
Smoking %	9.5%	17.1%	0.381
Alcoholism %	0%	2.6%	**<0.001**

The data are expressed as a mean ± SD (Student’s *t*-test) or as a percentage (chi2 test). BMI: Body mass index; HDL-C: high-density lipoprotein cholesterol; LDL-C: low-density lipoprotein cholesterol; SBP: systolic blood pressure; DBP: diastolic blood pressure; CF: cardiac frequency.

**Table 2 ijms-24-08641-t002:** Association between clinical parameters and miR/NOS3 expression with risk of developing CAD.

	χ^2^	OR	CI 95%	*p*
Age	6.177	1.088	1.018–1.163	**0.013**
HDL-C	8.863	0.848	0.761–0.945	**0.003**
SBP	5.951	1.071	1.014–1.133	**0.015**
miR-21-5p	8.411	8.373	1.992–35.206	**0.004**
miR-221-5p	6.839	6.447	1.595–26.061	**0.009**
*NOS3*	23.647	19.61	5.920–66.670	**<0.001**

χ^2^: chi^2^ value; OR: odds ratio; CI: confidence interval (binary logistic regression analysis).

**Table 3 ijms-24-08641-t003:** Comparison between controls and CAD patients either unmedicated or medicated with metformin regarding expression of the study miRs/NOS3.

	Control	CAD Unmedicated Group with Metformin	p1	CAD Medicated Group with Metformin	p2	p3
	*n* = 50	*n* = 48		*n* = 12		
**NOS3**	108.44 (5.70–1125.22)	25.11 (1.00–129.55)	**0.001**	23.98 (2.27–83.71)	**0.001**	0.776
**miR-21-5p**	7.54 (0.69–61.42)	15.43 (1.45–82.65)	**0.001**	7.35 (2.50–18.62)	0.983	0.022
**miR-221-5p**	7.47 (0.12–74.64)	21.79 (2.48–116.42)	**<0.001**	11.01 (2.78–46.85)	0.321	0.086
**miR-155-5p**	14.04 (0.74–78.72)	7.98 (0.91–33.15)	**0.049**	4.26 (2.63–21.63)	**0.050**	0.528

The data are expressed as median (min-max). p1 = CAD unmedicated group with metformin vs. healthy controls, p2 = CAD medicated group with metformin vs. control, *p* = 3 CAD unmedicated group with metformin vs. CAD medicated group with metformin (Kruskal-Wallis test).

## Data Availability

Data repository Figshare DOI 10.6084/m9.figshare.21295830.

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
