# Peer review of "Overexpression of microRNA-21-5p and microRNA-221-5p in Monocytes Increases the Risk of Developing Coronary Artery Disease"

_ijms, 2023, doi:10.3390/ijms24108641_

Round 1

Reviewer 1 Report

Summary:

In this paper, authors recruited 110 patients from National Cardiology Institute, and studied miR-221-5p, miR-21-5p, and miR-155-5p expression in monocytes. Authors linked miRNA expression level to the risk of Coronary arterial disease (CAD). How miRNA level regulate mRNA expression and link to specific disease onsite is a very interesting topic. However, this paper lack critical information to generate appropriate conclusions, and thus need major revisions:

First, the authors failed to provide the health condition for the control group. Are they healthy adults? Do they have other disease or even heart disease other than CAD (since all of them are recruited from National Cardiology Institute)? Is it possible that the control group has certain disease that potentially also affect miRNA level in monocytes which make them inappropriate to serve as controls?

Second, the authors mentioned most of the CAD patients in the patient group also have other complications such as type 2 diabetes and arterial hypertension and are on various medications. Since all the data points are analyzed in a pool and the authors didn’t distinguish different patients patient complications (e.g. CAD only without medication, CAD only with medication, CAD+ diabetes without medication, CAD+ diabetes with CAD medication, CAD + diabetes with diabetes medication, etc.....), is it possible the observed differences between miRNA expression level could also be affected by complications caused by other disease type or medications? In order to draw convincing conclusions, authors should consider include a much larger population of patients and deconvolute the data by disease type and medication.

More detailed comments please see below.

Detailed comments:

1.      Page 1 line 24, define CAD

2.      Page 1 line 29, describe “control group” since this is the first time this term appear and seems to be different from hypoglycemic agents group. What’s the difference between control group and unmedicated group?

3.      Page 1 line 33, “The metformin down regulate expression of miR-21-5p and miR-221-5p.” incomplete sentence

4.      Page 1 line34, define NOS3

5.      Table 1, column title missing

6.      Page 2 line 96: what is the criteria for control group? Are they also patients? What disease type if not CAD? Could their disease type affect miRNA expression level?

7.      Page 4, Figure 1: Data range of control group and CAD group have major overlap with only a few significantly higher values in CAD group (1A and 1B) or control group (1C and 1D). Could it be possible that those significantly different values come from patients with specific treatment or disease complication?

8.      Page 6 line 190: The authors claim that a large percentage of the CAD group patients also have type 2 diabetes and arterial hypertension and are thus treated by medication. So, it would be hard to deconvolute the effect from diabetes, arterial hypertension, and medication with the effect from CAD on miRNA expression and NOS3 level. How can the authors then conclude the observations you made are purely associated with CAD and not other complications?

Author Response

Thank you for your comments and suggestions

Changes and corrections are marked in red also in the manuscript.

 REVIEWER 1

Summary:

In this paper, authors recruited 110 patients from National Cardiology Institute, and studied miR-221-5p, miR-21-5p, and miR-155-5p expression in monocytes. Authors linked miRNA expression level to the risk of Coronary arterial disease (CAD). How miRNA level regulate mRNA expression and link to specific disease onsite is a very interesting topic. However, this paper lack critical information to generate appropriate conclusions, and thus need major revisions:

First, the authors failed to provide the health condition for the control group.

Answer: Page 8 line 281. It was added: “The criteria for the control group were not presenting any comorbidity”, and to ensure that they did not have atheroma or subclinical atherosclerosis, carotid intima-media thickness (cIMT) was evaluated by ultrasonography. All Participants answered standardized and validated questionnaires to obtain information on family and medical history, alcohol and tobacco consumption, and physical activity). Also, the clinical characteristics are shown in table 1

Are they healthy adults?

Answer: Yes, the criteria for the control group were not presenting any comorbidity. age: 48.67 ± 6.47 (controls) vs 62.10 ± 11.75 (CAD) patients (table 1)

Do they have other disease or even heart disease other than CAD (since all of them are recruited from National Cardiology Institute)? Is it possible that the control group has certain disease that potentially also affect miRNA level in monocytes which make them inappropriate to serve as controls?

Answer: Controls subjects were apparently healthy asymptomatic individuals without premature cardiovascular disease, recruited from blood bank donors and through brochures posted in Social Services centers.

Second, the authors mentioned most of the CAD patients in the patient group also have other complications such as type 2 diabetes and arterial hypertension and are on various medications. Since all the data points are analyzed in a pool and the authors didn’t distinguish different patients patient complications (e.g. CAD only without medication, CAD only with medication, CAD+ diabetes without medication, CAD+ diabetes with CAD medication, CAD + diabetes with diabetes medication, etc.....), is it possible the observed differences between miRNA expression level could also be affected by complications caused by other disease type or medications?

In order to draw convincing conclusions, authors should consider include a much larger population of patients and deconvolute the data by disease type and medication.

Answer: The sample size was calculated as shown on page 9 line 355 (Sample size). We agree with the reviewer's suggestion to stratify the patients according to other comorbidities, the correlation according to the drug was performed (miR/NOS3 expression with the use of drugs: (page 5, line 143-155)).

More detailed comments please see below.

Detailed comments:

  1. Page 1 line 24, define CAD

Answer: The definition was added. Coronary artery disease

  1. Page 1 line 29, describe “control group” since this is the first time this term appears and seems to be different from hypoglycemic agents group. What’s the difference between control group and unmedicated group?

Answer. Your observation is correct. This was already corrected. “the comparison was made between the group with CAD not medicated with metformin versus the healthy control group and group with CAD medicated with metformin”.

“. The results show significant increases in miR-21-5p in CAD unmedicated group with metformin patients vs. healthy control group (p= 0.001) and vs. CAD medicated group with metformin (p = 0.022).”

This analysis was carried out since a high percentage of the patients were treated with this hypoglycemic agent.

  1. Page 1 line 33, “The metformin down regulate expression of miR-21-5p and miR-221-5p.” incomplete sentence

Answer: It was already corrected. “In addition, in the CAD group, the metformin downregulated the expression of miR-21-5p and miR-221-5p”

  1. Page 1 line34, define NOS3

Answer. The definition was added: nitric oxide synthase 3 (NOS3)

  1. Table 1, column title missing.

Answer: Is correct, there was an error inserting the table. It was already corrected

  1. Page 2 line 96: what is the criteria for control group? Are they also patients? What disease type if not CAD?

Answer: This is mentioned in the methodology

Page 8 line 281 (To ensure that the control group did not present atheroma or subclinical atherosclerosis, carotid intima-media thickness (cIMT) was evaluated by ultrasonography. All Participants answered standardized and validated questionnaires to obtain information on family and medical history, alcohol and tobacco consumption, and physical activity).

Table 1 also shows the clinical characteristics

Could their disease type affect miRNA expression level?

Answer: In the control group there were no patients with arterial hypertension or diabetes. (Table 1).

  1. Page 4, Figure 1: Data range of control group and CAD group have major overlap with only a few significantly higher values in CAD group (1A and 1B) or control group (1C and 1D). Could it be possible that those significantly different values come from patients with specific treatment or disease complication?

Answer: Your question is correct. The recruited patients were not selected for treatment or complications of the disease. The treatment they received varied according to medical criteria, however the drugs consumed were statins, hypoglycemic and hypertensive drugs. Therefore, the analysis shown in table 2 was carried out. Of the main risk factors for the disease and table 3, analyzed by the use of metformin.

On the other hand, the sample size of patients with CAD without medication would have to be increased (which is difficult since they would have to be recruited at the onset of the disease) and thus establish a cut-off point to determine from which expression level, is considered a risk factor for developing CAD.

However, these differences in expression could be due to complications in the disease. Therefore, later we would have to do a prospective study:

1) To determine if in the case of healthy controls that presented a greater increase in the expression of miRNA-21-5p and miRNA-221-5p, they will develop CAD in the following years. And, conversely, to know if those healthy subjects who presented a greater increase in the expression of miRNA-155-5p and NOS3, will not present the disease (protective effect).

2) To determine whether, in the case of patients with CAD who presented an increase in the expression of miRNA-21-5p and miRNA-221-5p, whether they will have complications due to the disease or a poor prognosis.

  1. Page 6 line 190: The authors claim that a large percentage of the CAD group patients also have type 2 diabetes and arterial hypertension and are thus treated by medication. So, it would be hard to deconvolute the effect from diabetes, arterial hypertension, and medication with the effect from CAD on miRNA expression and NOS3 level. How can the authors then conclude the observations you made are purely associated with CAD and not other complications?

Answer: This questioning is correct, however it is difficult to know the effect solely due to CAD without drugs; since the patients admitted to hospitalization, are already previously medicated from the beginning of the detection of the disease by the treating physician. Which makes it very difficult to be able to make this analysis and therefore a conclusion as proposed by the reviewer.

It is true that there are some confounding variables, however, the purpose of having performed a binary logistic regression was to determine if indeed the increase or decrease in the expression of our study miRNAs and NOS3 are directly associated with the disease and not with other complications.

The objective of binary logistic regression is to verify causal relationships between a qualitative dependent variable, in this case the EAC, and one or more independent variables. Within our model, we introduced all those variables that presented significant differences between healthy controls and patients with CAD, and that, in addition, could influence the expression levels observed in miRNAs and NOS3. Our results indicated that the expression of these is directly associated with the disease and that, in addition, their deregulation, either up or down, is a risk factor for developing CAD (Table 2). If the deregulation in the expression of these were due to other comorbidities, in this case type 2 diabetes mellitus or hypertension, the expression of the miRNAs and NOS3 would not have given us a significant result in the regression, and this would have indicated that the deregulation observed in its expression was not directly associated with CAD, but rather with one of these comorbidities or other confounding variables.

Reviewer 2 Report

Authors describe the possible correlation between miR-21-5p, -221-5p expression and NOS3 downregulation in monocytes in CAD development.

Is there evidence that miR-21 and -221 target NOS3 mRNA? Is there any previous report about miR-21, -221 interaction with NOS3 mRNA? I do not see any description about this interaction in either Introduction or Discussion. Presumption derived from web-based Target scan etc. is not the evidence about a miRNA interaction with specific mRNA.

Line 116, miR-21, -221 target NOS3 gene?? Not NOS3 mRNA???

Line 206, Downregulation of miR-155-5p in CAD tissue does not necessarily mean its involvement in CAD development.

Discussion is too long. Shorten it.

Author Response

Thank you for your comments and suggestions

Changes and corrections are marked in red also in the manuscript.

REVIEWER 2

 Authors describe the possible correlation between miR-21-5p, -221-5p expression, and NOS3 downregulation in monocytes in CAD development.

Is there evidence that miR-21 and -221 target NOS3 mRNA? Is there any previous report about miR-21, -221 interaction with NOS3 mRNA? I do not see any description about this interaction in either Introduction or Discussion. Presumption derived from web-based Target scan etc. is not the evidence about a miRNA interaction with specific mRNA.

Answer: This sentence was added “Previous studies are related to the expression of miRNA-21-5p and miRNA-221-5p with NOS3 in both animal models and in humans….

 Cerda, et al. in endothelial cells transfected showed a decrease in miR-221 after statin treatment correlated with the increase in NOS3 mRNA levels (51). On the other hand, the group of Suárez et al. they observed that the transfection of miRNAs mimics in endothelial cells decreases the protein levels of NOS3 (52). In the case of miRNA-21-5p, in hypertensive patients with high intima media thickness, found a significant increase in the expression of miRNA-21-5p correlated positively with CIMT and negatively with the expression of NOS3, and nitric oxide (48).”

Line 116, miR-21, -221 target NOS3 gene?? Not NOS3 mRNA???

Answer: It was modified.  “NOS3 is responsible for the synthesis of nitric oxide in blood vessels and its mRNA is a target of miR-21-5p, miR-155-5p, and miR-221-5p”. Since the binding of multiple miRNAs to the same mRNA through different target sites can have a large effect on gene function.

Line 206, Downregulation of miR-155-5p in CAD tissue does not necessarily mean its involvement in CAD development.

Answer: Some modifications were made in order to better explain that the decrease in the expression of miR-155 may participate in the development of the disease.

“It has been suggested that miR-155-5p expression may have different effects depending on the stage of atheromatous plaque development. Its downregulation could represent a feedback mechanism for controlling the over activation of immune cells. According to previous studies in a mouse model with atherosclerosis, the expression of miRNA-155-5p decreases in these cells in response to inflammation, and thus it could be associated with the formation of atherosclerotic plaque. However, it is necessary, for example, to carry out complementary studies in vivo to ensure that its inhibition in mice with atherosclerosis produces a decrease in plaque size. Therefore, it is necessary to carry out complementary studies in vivo to ensure that its inhibition produces a decrease in the size of the plaque.”.

Round 2

Reviewer 1 Report

1. Compared to the original version, in this resubmitted version, clarifications and edits have been made according to reviewer feedback.

2. Although the authors' responses to reviewer's major concerns (questions 7 and 8) didn't fully answer the questions due to limitations in patient recruiting, the authors' answers sound convincing enough under current resources. I suggest the authors insert your answers to question 7 and 8 in the result or discussion session of the main manuscript after proper rephrase, because those are all very good discussions and future readers may benefit from those discussions too.

In case you don't have access to the original letter, I copied them here for your reference: "

  1. Page 4, Figure 1: Data range of control group and CAD group have major overlap with only a few significantly higher values in CAD group (1A and 1B) or control group (1C and 1D). Could it be possible that those significantly different values come from patients with specific treatment or disease complication?

Answer: Your question is correct. The recruited patients were not selected for treatment or complications of the disease. The treatment they received varied according to medical criteria, however the drugs consumed were statins, hypoglycemic and hypertensive drugs. Therefore, the analysis shown in table 2 was carried out. Of the main risk factors for the disease and table 3, analyzed by the use of metformin.

On the other hand, the sample size of patients with CAD without medication would have to be increased (which is difficult since they would have to be recruited at the onset of the disease) and thus establish a cut-off point to determine from which expression level, is considered a risk factor for developing CAD.

However, these differences in expression could be due to complications in the disease. Therefore, later we would have to do a prospective study:

1) To determine if in the case of healthy controls that presented a greater increase in the expression of miRNA-21-5p and miRNA-221-5p, they will develop CAD in the following years. And, conversely, to know if those healthy subjects who presented a greater increase in the expression of miRNA-155-5p and NOS3, will not present the disease (protective effect).

2) To determine whether, in the case of patients with CAD who presented an increase in the expression of miRNA-21-5p and miRNA-221-5p, whether they will have complications due to the disease or a poor prognosis.

  1. Page 6 line 190: The authors claim that a large percentage of the CAD group patients also have type 2 diabetes and arterial hypertension and are thus treated by medication. So, it would be hard to deconvolute the effect from diabetes, arterial hypertension, and medication with the effect from CAD on miRNA expression and NOS3 level. How can the authors then conclude the observations you made are purely associated with CAD and not other complications?

Answer: This questioning is correct, however it is difficult to know the effect solely due to CAD without drugs; since the patients admitted to hospitalization, are already previously medicated from the beginning of the detection of the disease by the treating physician. Which makes it very difficult to be able to make this analysis and therefore a conclusion as proposed by the reviewer.

It is true that there are some confounding variables, however, the purpose of having performed a binary logistic regression was to determine if indeed the increase or decrease in the expression of our study miRNAs and NOS3 are directly associated with the disease and not with other complications.

The objective of binary logistic regression is to verify causal relationships between a qualitative dependent variable, in this case the EAC, and one or more independent variables. Within our model, we introduced all those variables that presented significant differences between healthy controls and patients with CAD, and that, in addition, could influence the expression levels observed in miRNAs and NOS3. Our results indicated that the expression of these is directly associated with the disease and that, in addition, their deregulation, either up or down, is a risk factor for developing CAD (Table 2). If the deregulation in the expression of these were due to other comorbidities, in this case type 2 diabetes mellitus or hypertension, the expression of the miRNAs and NOS3 would not have given us a significant result in the regression, and this would have indicated that the deregulation observed in its expression was not directly associated with CAD, but rather with one of these comorbidities or other confounding variables."

3. I still suggest further analyze your current data shown in Figure 1. Can you add additional figures or tables or a few sentences in discussion showing miR-221-5p, miR-21-5p, miR-155-5p and NOS3 level in patients grouped by different health conditions, complications and medications? e.g. Patients with only CAD, patients with CAD+ diabetes, patients with CAD + arterial hypertension, with and without medication? Although the number of patients might be limited in each group, the authors might still be able to see some trend or get more clarifications on how disease complications and medication condition affect miRNA expression (or make your current conclusions more convicting by showing disease complication and medication are not associated with miRNA level). 

Author Response

Thank you for your comments and suggestions

Changes and corrections are marked in red also in the manuscript. 

REVIEWER 1

Although the authors' responses to reviewer's major concerns (questions 7 and 8) didn't fully answer the questions due to limitations in patient recruiting, the authors' answers sound convincing enough under current resources. I suggest the authors insert your answers to question 7 and 8 in the result or discussion session of the main manuscript after proper rephrase, because those are all very good discussions and future readers may benefit from those discussions too.

  1. Page 4, Figure 1: Data range of control group and CAD group have major overlap with only a few significantly higher values in CAD group (1A and 1B) or control group (1C and 1D). Could it be possible that those significantly different values come from patients with specific treatment or disease complication?
  2. Page 6 line 190: The authors claim that a large percentage of the CAD group patients also have type 2 diabetes and arterial hypertension and are thus treated by medication. So, it would be hard to deconvolute the effect from diabetes, arterial hypertension, and medication with the effect from CAD on miRNA expression and NOS3 level. How can the authors then conclude the observations you made are purely associated with CAD and not other complications?

Answer: the answer was inserted in the discussion (Page  6 Line 205):

Therefore, to verify that the expression of these miR is directly associated with CAD, the binary logistic regression analysis was performed. It was confirmed that their deregulation, either up or down, is a risk factor for developing the disease (as shown in the Table 2). If the deregulation in the expression of these were due to other comorbidities, in this case type 2 diabetes mellitus or hypertension, the expression of the miRNAs and NOS3 would not have given us a significant result in the regression, and this would have indicated that the deregulation observed in its expression was not directly associated with CAD, but rather with one of these comorbidities or other confounding variables.

However, is hard to deconvolute the effect of diabetes, arterial hypertension, and medication with the effect of CAD on miRNA expression and NOS3 level. Therefore, it is necessary to take into count that is difficult to capture patients with CAD without any treatment, so it is necessary to try to increase the sample size of this type of patient. A follow-up study could be done in our study subjects to 1) Determine if healthy controls that presented a greater increase in the expression of miRNA-21-5p and miRNA-221-5p; will develop CAD in the following years. Conversely, to know if those healthy subjects who presented a greater increase in the expression of miR-NA-155-5p and NOS3, will not present the disease (protective effect). 2) To determine if, in the case of patients with CAD who presented increased expression of miRNA-21-5p and miRNA-221-5p, they will have complications from the disease or a poor prognosis.

I still suggest further analyze your current data shown in Figure 1. Can you add additional figures or tables or a few sentences in discussion showing miR-221-5p, miR-21-5p, miR-155-5p and NOS3 level in patients grouped by different health conditions, complications and medications? e.g. Patients with only CAD, patients with CAD+ diabetes, patients with CAD + arterial hypertension, with and without medication? Although the number of patients might be limited in each group, the authors might still be able to see some trend or get more clarifications on how disease complications and medication condition affect miRNA expression (or make your current conclusions more convicting by showing disease complication and medication are not associated with miRNA level). 

Answer: You are right in your argument.

Page 5 line 164.  Finally, with respect to miR-221-5p, miR-21-5p, miR-155-5p and, NOS3 expression grouped by T2DM or consumption of statins and antihypertensive drugs, no significant differences were found (data not shown).

Page 7, line 240.  With respect to miR-221-5p, miR-21-5p, miR-155-5p, and NOS3 expression grouped in CAD patients and consumption different antihypertensive and lipid-lowering drugs, no significant differences were found. It could be probably it was due to here wide variety of treatments with antihypertensive (Diuretics, Beta-blockers, ACE inhibitors, Angiotensin II receptor blockers, Calcium channel blockers, Alpha blockers, Alpha-2 Receptor Agonists, Vasodilators ) and lipid-lowering ( statins , Fibrate,  ezetimibe ) drugs  in patients  with CAD

Reviewer 2 Report

Authors should insert proper references.

Line 118; its mRNA is a 118 target of miR-21-5p, miR-155-5p, and miR-221-5p (Ref. ???)

Line 119; Since the binding of multiple miRNAs to the same mRNA through different target sites can have a large effect on gene function. ???? – Are there previous reports that miR-21-5p, miR-155-5p, and miR-221-5p bind NOS3 mRNA at the same time or compete to bind the target sites? (Ref. ???)

Line 254; Add reference. (Ref. ???)

Author Response

Thank you for your comments and suggestions

Changes and corrections are marked in red also in the manuscript.

REVIEWER 2

Authors should insert proper references.

Line 118; its mRNA is a 118 target of miR-21-5p, miR-155-5p, and miR-221-5p (Ref. ???) 

The references were inserted in the proper order according to the text

  1. Cengiz, M, Yavuzer, S, Kılıçkıran, Avc, B, Yürüyen, M, Yavuzer, H, Dikici, SA, Karataş, ÖF, Özen, M, Uzun, H, Öngen, Z (2015). Circulating miR-21 and eNOS in subclinical atherosclerosis in patients with hypertension. Clin Exp Hypertens. 37(8): 643-649. doi: 10.3109/10641963.2015.1036064.
  1. Zhu, N, Zhang, D, Chen, S, Liu, X, Lin, L, Huang, X, Guo, Z, Liu, J, Wang, Y, Yuan, W, Qin, Y (2011). Endothelial enriched microRNAs regulate angiotensin II-induced endothelial inflammation and migration. Atherosclerosis . 2011 Apr;215(2):286-93. doi: 10.1016/j.atherosclerosis.2010.12.024.

Line 119; Since the binding of multiple miRNAs to the same mRNA through different target sites can have a large effect on gene function. ???? – Are there previous reports that miR-21-5p, miR-155-5p, and miR-221-5p bind NOS3 mRNA at the same time or compete to bind the target sites? (Ref. ???)

  1. Zhu, N, Zhang, D, Chen, S, Liu, X, Lin, L, Huang, X, Guo, Z, Liu, J, Wang, Y, Yuan, W, Qin, Y (2011). Endothelial enriched microRNAs regulate angiotensin II-induced endothelial inflammation and migration. Atherosclerosis . 2011 Apr;215(2):286-93. doi: 10.1016/j.atherosclerosis.2010.12.024.

Line 254; Add reference. (Ref. ???)

  1. Verjans R., Peters T., Beaumont F.J., van Leeuwen R., van Herwaarden T., Verhesen W., Munts C., Bijnen M., Henkens M., Diez J., et al. MicroRNA-221/222 Family Counteracts Myocardial Fibrosis in Pressure Overload-Induced Heart Failure. Hypertension. 2018;71:280–288. doi: 10.1161/HYPERTENSIONAHA.117.10094.
  1. Elton TS, Selemon H, Elton SM, Parinandi NL. Regulation of the MIR155 host gene in physiological and pathological processes. Gene 2013;532:1–12. doi: 10.1016/j.gene.2012.12.009.
